# Antibacterial Methyl Ester Cembranoids from the Soft Coral *Sarcophyton ehrenbergi* and Their Structural Elucidation

**DOI:** 10.3390/md23040170

**Published:** 2025-04-15

**Authors:** Meng-Jun Wu, Song-Wei Li, Fei Xu, Ming-Zhi Su, Yue-Wei Guo

**Affiliations:** 1Liangzhu Laboratory, Zhejiang University Medical Center, 1369 West Wenyi Road, Hangzhou 311121, China; mow@zju.edu.cn; 2School of Medicine, Shanghai University, Shanghai 200444, China; songweili@shu.edu.cn; 3Department of Gastroenterology of the Second Affiliated Hospital and Institute of Pharmaceutical Biotechnology, School of Medicine, Zhejiang University, Hangzhou 310058, China; fxu23@zju.edu.cn; 4Shandong Laboratory of Yantai Drug Discovery, Bohai Rim Advanced Research Institute for Drug Discovery, Yantai 264117, China; mzsu@baridd.ac.cn

**Keywords:** marine natural product, *Sarcophyton ehrenbergi*, cembranoid, absolute configuration, antibacterial activity

## Abstract

Six previously undescribed methyl ester cembranoids, namely sarcoehrenolides L–Q (**1**–**6**), along with three known related ones (**7**–**9**), were isolated from the soft coral *Sarcophyton ehrenbergi* collected off Weizhou Island in the South China Sea. Their structures and absolute configurations were unambiguously established in the light of extensive spectroscopic data analysis, X-ray diffraction analysis, chemical conversion, and TDDFT-ECD calculations. All isolated compounds were evaluated via in vitro bioassays to investigate their antibacterial activity against eighteen human and fish pathogens. Compounds **2**, **8**, and **9** exhibited moderate antibacterial activity against *Streptococcus parauberis* with MIC values of 38.8, 37.4 and 31.6 μg/mL, respectively.

## 1. Introduction

Since the 19th century, marine soft corals (phylum Cnidaria, class Anthozoa, subclass Octocoralia, order Alcyonaceae), as one of the typical marine invertebrates, encompassing families such as Alcyoniidae, Clavulariidea, Nephteheidae, and Xeniidae, etc. [1], have garnered significant research interest owing to the continuous discovery of structurally novel and biologically active secondary metabolites [2]. Scientists have conducted systematic chemical investigation of approximately one hundred species of soft corals, primarily focusing on genera such as *Cladiella*, *Lobophytum*, *Sarcophyton*, *Sinularia*, *Nephthea*, and *Xenia*, revealing a diverse array of bioactive secondary metabolites with potential pharmaceutical and ecological significance [3,4,5].

Diverse soft corals are capable of producing abundant and structurally novel secondary metabolites, including diterpenes, sesquiterpenes, steroids, and other bioactive compounds, which display remarkable chemical diversity in terms of structural types [6]. Analysis reveals that diterpenes represent the most abundant and structurally diverse class of secondary metabolites in soft corals, exhibiting the broadest spectrum of biological functions. The biosynthesis of diterpenes is based on a series of diterpene synthases, which act on the common biogenic precursor, geranylgeranyl pyrophosphate (GGPP), and the biosynthetic process emerged very early in the evolution of nature, making the natural world a huge reservoir of diterpenoid natural products [7]. Therefore, from both the biological and pharmacological perspectives, diterpenes are also the most interesting structural category. To date, over 30 distinct types of diterpene carbon skeletons have been isolated from soft corals, with the cembrane-type diterpenoids (cembranoids) being the most prevalent and representative marine-derived diterpene skeleton. Structurally, cembranoids are characterized by a 14-membered carbon ring and considered to be formed by a one-step cyclization reaction involving C-1 and C-14 of GGPP [8]. Subsequently, CYP450 enzymatic modifications lead to various structural transformations in cembranoids, including epoxidation of double bonds, oxidation of the carbon skeleton to form carbonyl, aldehyde, hydroxyl, acyl, and carboxyl functional groups, and further generate furan or lactone rings, which further enhance the structural complexity of cembranoids [9].

Our previous chemical investigations on the soft coral *S. ehrenbergi* have resulted in the isolation of a series of previously undescribed cembranoids, including sarcoehrenins A–J and sarcoehrenolides F–K, which exhibited TNF-*α* inhibitory and/or *α*-glucosidase inhibitory activities [10,11]. Inspired by these findings, we extended our chemical exploration of this species, leading to the discovery of six new 19-methyl ester cembranoids with different structural features, namely sarcoehrenolides L–Q (**1**–**6**), along with three known related analogues (**7**–**9**) (Figure 1). Herein, we report the isolation, structural elucidation, and biological evaluation of these new cembranoids, further expanding the chemical diversity and pharmacological potential of this unique marine natural product family.

## 2. Result and Discussion

Samples of *S. ehrenbergi* were frozen immediately at −20 °C after collection and stored at that temperature before they were exhaustively extracted by acetone. The Et_2_O-soluble portion of the acetone extract was subjected to repeated column chromatography (CC) (silica gel, Sephadex LH-20 and reversed-phase HPLC) to yield six new 19-methyl ester cembranoids sarcoehrenolides L–Q (**1**–**6**), along with three known analogues (**7**–**9**). The known compounds were rapidly characterized as xiguscabrate B [12], sarcoehrenin C [10], and xiguscabrate A [12], respectively, by a comparison of their NMR data with those reported in the literature.

All isolated compounds **1**–**9** showed characteristic NMR signals consistent with the cembrane-type diterpenoid framework, featuring a conjugated diene moiety. The structural novelty of new compounds **1**–**6**, compared to the co-occurring known compounds **7**–**9** [10,12] mainly manifested in their degree of oxidation and the specific substitution patterns of the acetoxy group. The geometry of double bonds Δ^1,2^ and Δ^3,4^ was assigned to the *E* configuration for all compounds, supported by the clear NOESY correlations of H-2/Me-16 and H-2/H-18. Furthermore, oxidation and esterification that occurred on C-19 in compounds **1**–**6** resulted in the formation of a 19-methyl ester, as confirmed by HMBC correlations from H-7 to C-19 and from OMe-3′ to C-19 (Figure 2). These structural features highlight the unique chemical modifications and diversity within this class of cembranoids.

Compound **1** was isolated as colorless crystals with a melting point of 118.2–119.0 °C. Its molecular formula, C_23_H_34_O_5_, was determined by the HR-ESIMS quasi-molecular ion peak at *m*/*z* 413.2305 ([M + Na]^+^, calcd. for C_23_H_34_O_5_Na, 413.2298), indicating seven degrees of unsaturation. The IR spectrum of **1** displayed absorption at 1740 cm^−1^, suggested the presence of a carbonyl group in its molecules. The ^1^H and ^13^C NMR data (Table 1 and Table 2) of **1** were reminiscent of the co-occurring compound **7**, xiguscabrate B, a known cembranoid previously isolated from the South China Sea soft coral *Sinularia scabra* [12]. Careful comparison of the NMR data of **1** and **7** revealed that the main difference between them happened at C-6. The ^13^C NMR chemical shift in C-6 in **1** was obviously downfield shifted (*δ*_C_ 27.0 in **7** and 71.5 in **1**), indicating that acetoxylation occurred at the C-6 of **1**, which was in agreement with the diagnostic HMBC cross-peaks of H-6 (*δ*_H_ 6.01) to C-1′ (*δ*_C_ 170.2) and the 42 mass units difference between their ESIMS data. Subsequently, a detailed analysis of the ^1^H–^1^H COSY and HMBC correlations established the planar structure of **1** as shown in Figure 2.

The geometry of double bond Δ^7,8^ of **1** was assigned to the *Z* configuration by the observed NOESY correlation of H-7/H_2_-9. The relative configuration of C-11/C-12 in **1** was suggested to be same *trans* (11*S**, 12*S**) as that of **7** on the basis of the highly similar ^13^C NMR data between **1** and **7**. However, determining the relative configuration between asymmetric centers C-6 and C-11/C-12 remains a challenge. To confirm the structure and stereochemistry of **1**, we attempted to obtain (and fortunately successfully obtained) suitable single crystals via recrystallization in methanol. The X-ray crystallographic analysis using Cu Kα radiation (λ = 1.54178 Å) firmly disclosed the planar structure of **1** and determined its absolute configuration to be 6*S*, 11*S*, 12*S* with a Flack parameter of −0.12 (8) (Figure 3).

Compound **2** was obtained as an optically active colorless oil. Its molecular formula, C_23_H_32_O_5_, consistent with eight degrees of unsaturation, was determined by the HR-ESIMS quasi-molecular ion peak at *m/z* 389.2316 ([M + H]^+^, calcd 389.2323, C_23_H_33_O_5_). The ^1^H and ^13^C NMR data of **2** (Table 1 and Table 2) showed high similarity to those of co-occurring known compound **8**, sarcoehrenin C [10], with the major difference observed at C-13. The main manifestation was the replacement of a methylene (C-13, *δ*_C_ 38.0) in **8** by a carbonyl (C-13, *δ*_C_ 200.6) in **2**, which was confirmed by the diagnostic HMBC cross-peaks of Me-20 (*δ*_H_ 1.72) to C-11 (*δ*_C_ 141.2)/C-12 (*δ*_C_ 137.1)/C-13 and H-11 (*δ*_H_ 6.59) to C-13 in **2**. Further 2D NMR spectra analysis allowed the unambiguous determination of the planar structure of compound **2**.

The geometry of double bonds Δ^7,8^ and Δ^11,12^ of **2** was assigned to *Z* and *E* configurations, respectively, on the basis of the obvious NOESY correlation of H-7/H_2_-9 and H_2_-10/Me-20 (Figure 2). The absolute configuration of asymmetric center C-6 in **2** was determined by TDDFT-ECD calculation. As shown in Figure 4, the Boltzmann-averaged ECD spectrum of the (6*S*)-**2** was well matched with the experimental one, while the calculated ECD spectrum of (6*R*)-**2** displayed an almost opposite curve (Figure 4). Therefore, compound **2** was elucidated to be the 13-carbonyl derivative of **8** and drawn as shown in Figure 1.

Compound **3**, an optically inactive colorless oil, had the molecular formula of C_21_H_30_O_3_ as established by the quasi-molecular ion peak at *m*/*z* 331.2266 ([M + H]^+^, calcd 331.2268, C_21_H_31_O_3_) in its HR-ESIMS spectrum, corresponding to 58 mass units less than that of **2**, suggesting that **3** has one less acetoxy group in its structure compared to **2**. The ^1^H–^1^H COSY of H_2_-5/H_2_-6/H-7 spin coupling system indicated the absence of an acetoxyl group at the C-6 position in compound **3**, which was in agreement with the 58 mass unit difference. In fact, due to the replacement of O-methine by methylene at C-6, the ^13^C NMR chemical shift in C-6 was, as expected, upfield shifted (*δ*_C_ 70.5 in **2** vs. *δ*_C_ 27.4 in **3**), whereas those of adjacent carbon atoms C-4, C-5, C-7 and C-8 were all reasonably upfield or downfield shifted (Table 2). The geometry of double bonds Δ^7,8^ and Δ^11,12^ was suggested to involve *Z* and *E* configurations, respectively, according to the NOESY correlations of H-7/H_2_-9, and H_2_-10/Me-20 (Figure 2). Herein, the structure of **3** was defined as shown in Figure 1.

Compound **4** was isolated as a colorless oil with the molecular formula C_21_H_30_O_4_ as deduced from the HR-ESIMS ion peak at *m*/*z* 369.2039 ([M + Na]^+^, calcd 369.2036, C_21_H_30_O_4_Na), implying 16 mass units (oxygen atom) more than that of **3**. The 1D NMR spectra of **4** showed high similarity to those of **3**. A careful comparison of the ^1^H and ^13^C NMR data of **4** and **3** revealed that the differences between them mainly happened at H-11, Me-20, C-11, C-12 and C-13, suggesting the existence of an epoxy group at C-11 (*δ*_C_ 59.6) and C-12 (*δ*_C_ 64.4) in **4** instead of a trisubstituted double bond in **3**, in agreement with the 16 mass units difference. This inference was further confirmed by the HMBC cross-peaks from Me-20 (*δ*_H_ 1.42) to C-11/C-12. Accordingly, compound **4** was identified as the epoxidized derivative of **8** at double bond Δ^11,12^. The geometry of double bond Δ^7,8^ of **4** was assigned to the *Z* configuration by the observed NOESY correlation of H-7/H_2_-9. The relative configuration of C-11/C-12 in **4** was tentatively assigned to be 11*S** and 12*R** based on the absence of NOESY correlation between H-11 (*δ*_H_ 3.19) and Me-20 (*δ*_H_ 1.42). In this case, the absolute configuration of **4** was also determined by the TDDFT/ECD calculation. The Boltzmann-averaged ECD curve of (11*S*, 12*R*)-**4** was closely matched to the experimental ECD spectrum (Figure 4). Consequently, the absolute configuration of **4** was determined to be 11*S*, 12*R*.

Compound **5** has a molecular formula of C_21_H_32_O_3_, as determined by the HR-ESIMS ion peak at *m*/*z* 355.2240 ([M + Na]^+^, calcd 355.2244, C_21_H_32_O_3_Na), implying 6 degrees of unsaturation. The ^1^H and ^13^C NMR data of **5** were similar to those of **3** and co-occurring compound **9**, xiguscabrate A [12], but the main difference lay in the functional group at C-13. In fact, the C-13 position in compound **5** was replaced by a hydroxyl group instead of the ketone carbonyl group in **3**. Further 2D NMR spectroscopic analysis (Figure 2), including ^1^H–^1^H COSY and HMBC, confirmed the planar structure of **5**. For further confirmation of the absolute configuration of **5**, the TDDFT/ECD calculation was performed again. Finally, the calculated ECD spectrum of (13*R*)-**5** appeared to be highly similar to the experimental one (Figure 4). Therefore, the absolute configuration of **5** was established to be 13*R*.

Compound **6** was obtained as a colorless oil with the molecular formula of C_23_H_34_O_4_ as established by the HR-ESIMS ion peak at *m*/*z* 397.2355 ([M + Na]^+^, calcd 397.2349, C_23_H_34_O_4_Na), implying 42 mass units more than that of **5**. Careful analysis revealed that the 1D NMR spectroscopic features of **6** (Table 1 and Table 2) closely resembled those of **5**, indicating a high degree of structural similarity between them. The sole structural difference lay at the C-13 position, where the hydroxyl group in **5** was replaced by an acetoxyl group (*δ*_H_ 2.03; *δ*_C_ 21.6, 170.4) in **6**, consistent with the observed 42 mass unit difference between the two compounds. Due to the acetylation, the chemical shift in C-13 was obviously downfield shifted from *δ*_C_ 79.0 in **5** to *δ*_C_ 80.5 in **6**. A detailed 2D NMR analysis further confirmed the planar structure of **6**. In order to confirm the structural relationship between compounds **5** and **6**, an acetylation reaction was carried out on **5**. The acetylated product **5a** was successfully obtained by treating **5** with acetic anhydride in dry pyridine for 1 h at room temperature, whose NMR spectrum (Appendix A) and specific optical value were in full agreement with those of naturally isolated compound **6**. Thus, the structure of **6** was determined to be the 13-acetylation derivative of **5**.

As demonstrated in previous studies by our research group, cembranoids have exhibited significant activity across various pharmacological models. In line with these findings, the isolated compounds **1**–**9** were subjected to multiple pharmacological evaluations, including anticancer, anti-inflammatory, antibacterial, and neuroprotective assessments. In antibacterial bioassays against eighteen human and marine fish pathogens [13] (Appendix A), compounds **2**, **8** and **9** exhibited moderate antibacterial activity against *Streptococcus parauberis* with MIC values of 38.8, 37.4 and 31.6 μg/mL, respectively (Table 3), which are comparable to those of three previously reported cembranoids (lobocaloid B, 8.7 μg/mL; 11,12-epoxy-1*E*,3*E*,7*E*-cembratrien-15-ol, 30.4 μg/mL; sarcophytrol L, 32.2 μg/mL) [14]. Additionally, compounds **1**–**9** were assessed for their neuroprotective potential. However, none exhibited significant anti-inflammatory activity in terms of inhibiting NO production in LPS-induced BV2 microglial cells. Further biological evaluations, including cytotoxicity assays, PPAR activation assays, and antioxidant activity assessments, are currently underway to explore additional pharmacological properties of these compounds.

Although the secondary metabolites, including structurally diverse cembrane-type diterpenes, isolated from marine soft corals exhibit enormous medicinal potential, several challenges remain that require further research and exploration [15]. A major limitation is the insufficient quantity of newly isolated compounds, which hinders in-depth studies on their mechanisms of action and structure–activity relationships (SAR). This has long been a bottleneck in marine natural products research. However, advancements in biotechnology have begun to address these challenges [16,17]. Recent studies have reported the identification of biosynthetic genes responsible for the production of defensive substances in soft corals [18,19,20,21]. Therefore, the application of synthetic biology and metabolic engineering offers a promising strategy to overcome supply limitations. By transferring biosynthetic gene clusters into heterologous hosts, such as bacteria *Escherichia coli* or fungi *Aspergillus oryzae*, researchers can produce diterpenes in a controlled and scalable manner [22]. Similar strategies could be employed for marine-derived diterpenes, enabling the large-scale production of these compounds for further research and development. These findings not only demonstrate that marine soft corals can autonomously synthesize important defensive small molecules but also offer reliable research protocols for the biosynthetic production of marine drug precursors, which could pave the way for overcoming supply limitations and accelerating the development of marine drugs.

## 3. Materials and Methods

### 3.1. General Experimental Procedures

Melting points were measured with an X-4 digital micro-melting-point apparatus. The X-ray measurement was made using a Bruker D8 Venture X-ray diffractometer with Cu Kα radiation (Bruker Biospin AG, Fällanden, Germany). Optical rotations were measured with a Perkinelmer 241 MC polarimeter (PerkinElmer, Fremont, CA, USA). The IR spectrum was recorded with a Nicolet 6700 spectrometer (Thermo Fisher Scientific, Waltham, MA, USA). ^1^H and ^13^C NMR spectra were acquired using a Bruker DRX-600 spectrometer (Bruker Biospin AG, Fällanden, Germany). The HR-ESI-MS spectra were recorded with a ZenoTOF7600 mass spectrometer (SCIEX, Framingham, MA, USA). Reversed phase (RP) HPLC was performed on an Agilent 1260 series liquid chromatograph, equipped with a DAD G1315D detector, at 210 nm (Agilent, Santa Clara, CA, USA). Agilent semi-preparative XDB-C18 column (5 μm, 250 × 9.4 mm) was employed for the purification.

### 3.2. Animal Materials

The soft coral *S. ehrenbergi* was collected from Weizhou Island (20°54′−21°10′ N, 109°00′–109°15′ E), the Guangxi Zhuang Autonomous Region, China, in May 2007, at a depth of –20 m, and identified by Professor Xiu-Bao Li from Hainan University. A voucher specimen (No. WZ-14) is available for inspection at the Shandong Laboratory of Yantai Drug Discovery.

### 3.3. Extraction and Isolation

The frozen animals (428.0 g, dry weight) were cut into small pieces and exhaustively extracted with acetone (3 × 1.5 L) using an ultrasonic bath for 15 min at room temperature. The combined organic extract was evaporated under reduced pressure, yielding a brown residue that was partitioned between Et_2_O and H_2_O. The Et_2_O layer was concentrated to obtain a dark brown residue (18.8 g), which was subjected to gradient silica gel (200–300 mesh) column chromatography (CC) with a mobile phase of 0–100% Et_2_O in petroleum ether (PE), resulting in six fractions (A–F). Fraction B (1131.7 mg) was first chromatographed over a Sephadex LH-20 column with PE/DCM/MeOH (2:1:1, *v*/*v*/*v*) to afford three subfractions (B1–B3). Subfraction B1 was further separated by silica gel CC (20 cm × 2 cm, 300–400 mesh) using PE–Et_2_O (10:1, *v*/*v*) as the eluent, yielding subfractions B1a–B1d. Subfraction B1a was purified via semi-preparative RP-HPLC (CH_3_CN–H_2_O, 90:10, *v*/*v*) to furnish compound **4** (2.2 mg, t_R_ = 15.9 min), **7** (5.6 mg, t_R_ = 18.3 min) and **9** (2.5 mg, t_R_ = 21.5 min). Fraction D (633.3 mg) was similarly processed through a Sephadex LH-20 column to yield subfractions D1–D3. Subfraction D2 was subjected to silica gel CC (300–400 mesh) with PE–Et_2_O (8:2, *v*/*v*), producing subfractions D2a–D2b. Subfraction D2a was further purified by semi-preparative RP-HPLC (CH_3_CN–H_2_O, 80:20, *v*/*v*), affording compounds **1** (2.0 mg, t_R_ = 15.8 min), **2** (3.2 mg, t_R_ = 10.5 min), and **5** (1.8 mg, t_R_ = 14.5 min). Fraction E (256.4 mg) underwent Sephadex LH-20 chromatography with DCM/MeOH (1:1, *v*/*v*) to yield subfractions E1–E2. Subfraction E1 was further separated by silica gel CC (300–400 mesh) with PE–Et_2_O (8:2, *v*/*v*), generating subfractions E1a–E1b. Subfraction E1a was purified via semi-preparative RP-HPLC (CH_3_CN–H_2_O, 70:30, *v*/*v*), yielding compounds **3** (2.9 mg, t_R_ = 8.8 min), **6** (1.6 mg, t_R_ = 10.7 min), and **8** (5.7 mg, t_R_ = 14.7 min).

### 3.4. Spectroscopic Data of Compounds

Sarcoehrenolide L (**1**): Colorless crystal, (m. p. 118.2–119.0 °C); ${\left[\alpha \right]}_{D}^{20}$ +55.4 (*c* 0.20, MeOH); IR (KBr) ν_max_ 2957, 2928, 2870, 1740, 1720, 1439, 1370, 1240, 1139, 1021 cm^−1^; UV (MeOH) *λ*_max_(log*ε*) 244.0 (3.5) nm; CD (MeOH) *λ* (Δ*ε*) 213.5 (+21.1), 206.0 (−5.2). For ^1^H and ^13^C NMR spectroscopic data, see Table 1 and Table 2; HR-ESIMS *m*/*z* 413.2305 ([M + Na]^+^; calcd. for C_23_H_34_O_5_Na, 413.2298).

Sarcoehrenolide M (**2**): Colorless oil; ${\left[\alpha \right]}_{D}^{20}$ +9.1 (*c* 0.15, MeOH); IR (KBr) ν_max_ 2958, 2927, 1739, 1720, 1661, 1436, 1369, 1235, 1203, 1020 cm^−1^; UV (MeOH) *λ*_max_(log*ε*) 232.8 (3.5) nm; CD (MeOH) *λ* (Δ*ε*) 208.6 (+14.1), 232.8 (−17.7). For ^1^H and ^13^C NMR spectroscopic data, see Table 1 and Table 2; HR-ESIMS *m*/*z* 389.2316 ([M + H]^+^; calcd. for C_23_H_33_O_5_, 389.2323).

Sarcoehrenolide N (**3**): Colorless oil; IR (KBr) ν_max_ 2926, 2856, 1659, 1440, 1378, 1199, 1133, 1083 cm^−1^. For ^1^H and ^13^C NMR spectroscopic data, see Table 1 and Table 2; HR-ESIMS *m*/*z* 331.2266 ([M + H]^+^; calcd. for C_21_H_31_O_3_, 331.2268).

Sarcoehrenolide O (**4**): Colorless oil, ${\left[\alpha \right]}_{D}^{20}$ +10.3 (*c* 0.22, MeOH); IR (KBr) ν_max_ 2957, 2925, 2870, 1702, 1438, 1379, 1247, 1196, 1131 cm^−1^; UV (MeOH) *λ*_max_(log*ε*) 242.5 (3.1) nm; CD (MeOH) *λ* (Δ*ε*) 218.0 (+7.0), 258.5 (+19.4), 318.0 (−7.1). For ^1^H and ^13^C NMR spectroscopic data, see Table 1 and Table 2; HR-ESIMS *m*/*z* 369.2039 ([M + Na]^+^; calcd. for C_21_H_30_O_4_Na, 369.2036).

Sarcoehrenolide P (**5**): Colorless oil, ${\left[\alpha \right]}_{D}^{20}$ +43.3 (*c* 0.05, MeOH); IR (KBr) ν_max_ 3500, 2957, 2925, 2870, 1716, 1436, 1379, 1246, 1208, 1018 cm^−1^; UV (MeOH) *λ*_max_(log*ε*) 245.0 (3.1) nm; CD (MeOH) *λ* (Δ*ε*) 217.5 (−7.2), 258.5 (+19.4), 316.5 (−6.8). For ^1^H and ^13^C NMR spectroscopic data, see Table 1 and Table 2; HR-ESIMS *m*/*z* 355.2240 ([M + Na]^+^; calcd. for C_21_H_32_O_3_Na, 355.2244).

Sarcoehrenolide Q (**6**): Colorless oil, ${\left[\alpha \right]}_{D}^{20}$ +19.6 (*c* 0.20, MeOH); IR (KBr) ν_max_ 2957, 2924, 2870, 1735, 1717, 1438, 1372, 1238, 1130, 1017 cm^−1^; UV (MeOH) *λ*_max_(log*ε*) 248.5 (3.3) nm; CD (MeOH) *λ* (Δ*ε*) 248.5 (+3.1). For ^1^H and ^13^C NMR spectroscopic data, see Table 1 and Table 2; HR-ESIMS *m*/*z* 397.2355 ([M + Na]^+^; calcd. for C_23_H_34_O_4_Na, 397.2349).

### 3.5. Calculation Section

Conformational searches using torsional sampling (MCMM) with the OPLS_2005 force field were performed via the Macromodel’s conformational search module, employing an energy window of 21 kJ/mol to generate initial conformers. Subsequently, the Boltzmann populations of these conformers were calculated based on their potential energies derived from the same force field, with only those exhibiting populations above 1% selected for further re-optimization. The re-optimization process and subsequent Time-Dependent Density Functional Theory (TDDFT) calculations were conducted using Gaussian 09 at the B3LYP/6-311G(d,p) level, incorporating the IEFPCM solvent model for acetonitrile. Frequency analyses were then carried out to confirm that the re-optimized geometries corresponded to the true energy minimum. Finally, the Boltzmann-averaged ECD spectra were generated and visualized using SpecDis1.62 software to facilitate comparison with experimental data.

### 3.6. Acetylation of Compound **5**

Compound **5** (1.0 mg) was dissolved in 2.0 mL of dry pyridine and mixed with 200 μL of acetic anhydride, and the mixtures were stirred at room temperature for one hour. The reactant was extracted with water and ether to obtain crude acetylated product, which was further purified by silica gel CC to afford a colorless oil compound **5a** (0.8 mg, 71% yield).

### 3.7. X-Ray Crystallographic Analysis for Compound **1**

Compound **1** was recrystallized in MeOH to obtain high-quality crystals. The crystallographic data were collected on a Bruker D8 Venture diffractometer equipped with Cu Kα radiation (λ = 1.54178 Å). The structures were solved with the ShelXT structure solution program using Intrinsic Phasing and refined with the ShelXL refinement package using Least Squares minimization.

Compound **1**: colorless crystals, T = 170 K, C_23_H_34_O_5_ (*M* = 390.50 g/mol), crystal size: 0.15 × 0.04 × 0.02 mm^3^, orthorhombic, space group P2_1_2_1_2_1_, *a* = 9.9161(2) A, *b* = 19.5833(4) A, *c* = 22.6521(5) A, *V* = 4398.81(16) A^3^, *Z* = 8, *μ* (Cu K*α*) = 0.657 mm^−1^, *D*_calc_ = 1.179 g/cm^3^, θ range = 5.966−149.590°, reflections collected 54106 [R_int_ = 0.0662], *R*_1_ = 0.0455 [I > 2σ(I)], *wR*_2_ = 0.1179 [all data], absolute structure parameter: −0.12(8).

### 3.8. Antibacterial Activity Assay

The human pathogens Staphylococcus aureus ATCC27154, Enterococcus faecium, Escherichia coli ATCC25922, Enterobacter cloacae ZR042, Enterobacter hormaechei 2R043, Pseudomonas aeruginosa ATCC10145, Pseudomonas aeruginosa 2200, Escherichia coli, Enterobacter cloacae, methicillin-resistant Staphylococcus aureus (MRSA), and Candida albicans ATCC76485 were donated by the Korea Institute of Science and Technology. The marine strains Streptococcus parauberis KSP28, Streptococcus parauberis SPOF3K, Lactococcus garvieae MP5245, Aeromonas salmonicida AS42, Phoyobacterium damselae FP2244, Pseudomonas fulva ZXM181, and Photobacterium halotolerans LMG22194T were provided by the National Fisheries Research &Development Institute, Korea. The MIC values of the compounds were determined using a modified 0.5 McFarland standard method. Two-fold serial dilutions of the compounds were prepared in DMSO. The turbidity of the bacterial suspensions was measured at 600 nm and adjusted with the medium to match the 0.5 McFarland standard (10^5^ colony-forming units/mL). Subsequently, 95 μL of bacterial culture was inoculated into each well of a 96-well plate, followed by the addition of 5 μL of the test solutions. The plates were then incubated at 37 °C for 12 h, and MIC values were determined in triplicate. To ensure that the vehicle had no significant effect on bacterial growth, each bacterial species was also cultured in LB broth containing DMSO at concentrations equivalent to those used in the test solutions.

## 4. Conclusions

In summary, through further chemical investigation of the soft coral *S. ehrenbergi* collected from the South China Sea, nine cembranoids were isolated and characterized from the organic extracts of this species using a combination of separation methods, including silica gel, Sephadex LH-20 column chromatography and RP-HPLC. Of these, six compounds were identified as previously unreported 19-methyl ester cembranoids. Furthermore, the structures, together with the absolute configurations of the six new compounds, were elucidated on the basis of various analytical methods, including NMR data comparison analysis, X-ray single-crystal diffraction analysis, chemical derivatization, and quantum mechanical calculations. This study expands the terpenoid family of marine invertebrates and enriches the diversity of marine natural products. Additionally, antibacterial activity screening of the isolated secondary metabolites revealed that compounds **2**, **8**, and **9** exhibited moderate activity against *Streptococcus parauberis*, with MIC values ranging from 31.6 to 38.8 μg/mL. Overall, the discovery of these structurally novel compounds enriches the chemical diversity of natural secondary metabolites and provides a valuable foundation for the research and development of innovative drugs.

## Figures and Tables

**Figure 1 marinedrugs-23-00170-f001:**
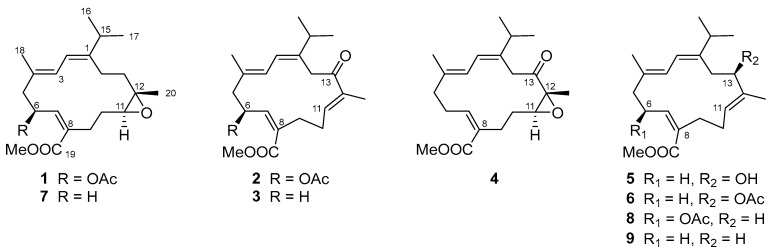
Chemical structures of compounds **1**–**9**.

**Figure 2 marinedrugs-23-00170-f002:**
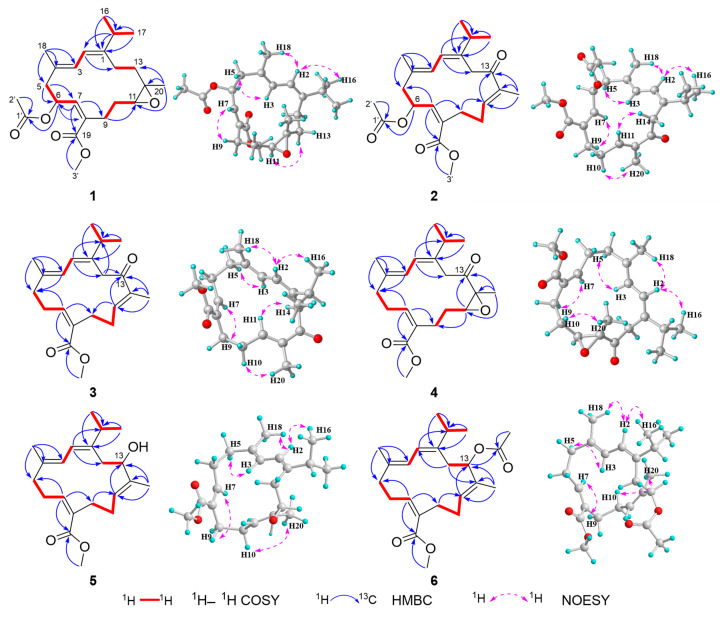
^1^H–^1^H COSY, selected key HMBC, and NOESY correlations of **1**–**6**.

**Figure 3 marinedrugs-23-00170-f003:**
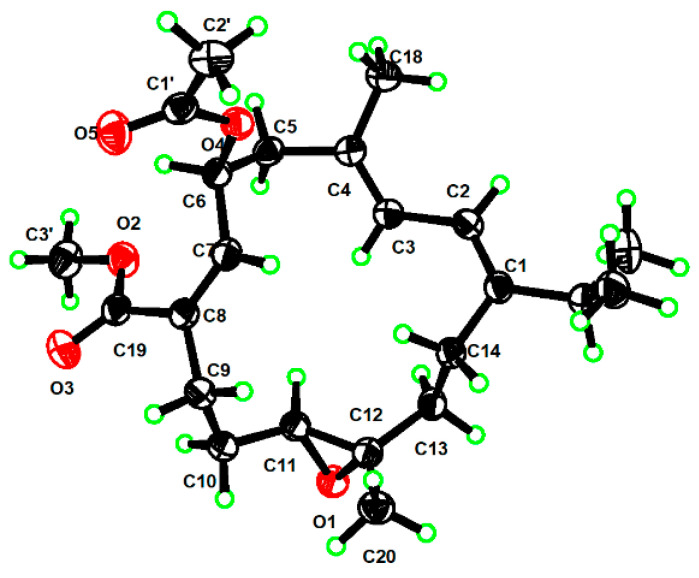
ORTEP drawing of **1** (the displacement ellipsoids are drawn at the 50% probability level).

**Figure 4 marinedrugs-23-00170-f004:**
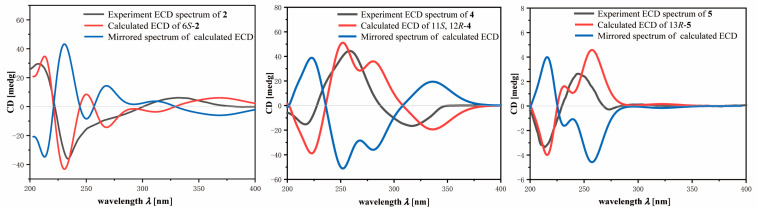
The assignment of the absolute configuration of **2**, **4** and **5** by comparing TDDFT-ECD calculated and experimental ECD spectra.

**Table 1 marinedrugs-23-00170-t001:** ^1^H NMR data for compounds **1**–**6** ^a^.

No.	*δ*_H_, Mult (*J* in Hz)					
	1	2	3	4	5	6
2	5.90, d (10.0)	6.06, d (11.3)	6.06, d (11.0)	6.06, d (10.9)	6.09, d (11.1)	6.09, d (11.2)
3	5.87, d (10.0)	6.13, d (11.3)	6.01, d (11.0)	5.84, d (10.9)	5.78, d (11.1)	5.80, d (11.2)
5	2.65, dd (14.2, 1.9)	2.60, dd (12.5, 3.6)	2.29, m	2.31, m	2.21, m	2.21, m
	2.53, dd (14.2, 6.2)	2.42, dd (12.5, 9.6)	2.29, m	2.25, m	2.21, m	2.21, m
6	6.01, m	6.08, dd (9.3, 3.6)	2.67, m	2.87, m	2.67, m	2.70, m
			2.67, m	2.48, m	2.62, m	2.59, m
7	5.98, dd (8.3, 1.3)	5.68, d (9.3)	5.83, t (7.8)	5.96, dd (9.4, 5.8)	5.70, t (6.8)	5.71, t (6.9)
9	2.91, d (14.1)	2.90, dd (12.4, 4.5)	2.44, m	2.51, m	2.54, dd (13.8, 7.7)	2.52, m
	2.08, m	2.11, d (12.4)	2.44, m	2.46, m	2.29, m	2.30, m
10	2.06, m	2.49, m	2.39, m	1.80, m	2.31, m	2.30, m
	1.48, m	2.23, m	2.39, m	1.67, m	2.16, m	2.15, m
11	2.84, dd (9.5, 3.9)	6.59, dd (8.8, 4.3)	6.66, t (6.7)	3.19, dd (7.7, 5.2)	5.25, t (6.9)	5.32, t (4.7)
13	2.05, m				3.96, br s	5.06, dd (6.7, 4.5)
	1.33, m					
14	2.06, m	3.86, d (12.8)	3.54, s	3.85, d (14.2)	2.74, dd (13.7, 6.4)	2.78, dd (13.9, 6.7)
	2.06, m	3.25, d (12.8)	3.54, s	2.62, d (14.2)	2.19, m	2.25, dd (13.9, 4.5)
15	2.32, m	2.29, m	2.32, m	2.51, m	2.43, m	2.37, m
16	1.05, d (6.8)	1.02, d (6.8)	1.02, d (6.8)	1.02, d (6.8)	1.10, d (6.8)	1.07, d (6.8)
17	1.05, d (6.8)	1.00, d (6.8)	1.02, d (6.8)	1.02, d (6.8)	1.08, d (6.8)	1.03, d (6.8)
18	1.79, s	1.90, s	1.78, s	1.75, s	1.75, s	1.75, s
20	1.26, s	1.72, s	1.72, s	1.42, s	1.61, s	1.57, s
OAc	2.03, s	2.04, s				2.03, s
OMe	3.81, s	3.79, s	3.74, s	3.75, s	3.75, s	3.74, s

^a^ Recorded in CDCl_3_ at 600 MHz. s (singlet), d (doublet), t (triplet), m (multiplet), and br s (broad singlet).

**Table 2 marinedrugs-23-00170-t002:** ^13^C NMR data for compounds **1**–**6** ^a^.

No.	*δ*_C_, Mult					
	1	2	3	4	5	6
1	147.8, C	142.3, C	141.4, C	139.9, C	145.0, C	143.7, C
2	118.6, CH	119.0, CH	119.3, CH	120.5, CH	119.9, CH	120.3, CH
3	124.8, CH	124.9, CH	121.9, CH	121.2, CH	121.3, CH	121.3, CH
4	132.4, C	133.3, C	136.9, C	137.5, C	135.3, C	135.7, C
5	43.2, CH_2_	45.3, CH_2_	39.8, CH_2_	39.6, CH_2_	38.8, CH_2_	38.9, CH_2_
6	71.5, CH	70.5, CH	27.4, CH_2_	27.5, CH_2_	26.7, CH_2_	26.7, CH_2_
7	140.4, CH	138.6, CH	144.2, CH	145.8, CH	143.7, CH	143.4, CH
8	132.5, C	134.1, C	130.5, C	129.6, C	130.1, C	130.2, C
9	32.2, CH_2_	33.8, CH_2_	33.8, CH_2_	32.3, CH_2_	33.8, CH_2_	33.5, CH_2_
10	25.9, CH_2_	26.5, CH_2_	27.0, CH_2_	25.6, CH_2_	25.3, CH_2_	25.2, CH_2_
11	60.6, CH	141.2, CH	141.9, CH	59.6, CH	126.2, CH	128.6, CH
12	61.6, C	137.1, C	137.0, C	64.4, C	139.0, C	134.4, C
13	38.3, CH_2_	200.6, C	201.1, C	208.6, C	79.0, CH	80.5, CH
14	26.1, CH_2_	40.8, CH_2_	40.3, CH_2_	36.2, CH_2_	36.7, CH_2_	34.5, CH_2_
15	35.1, CH	33.0, CH	33.2, CH	33.4, CH	35.3, CH	34.6, CH
16	22.2, CH_3_	21.8, CH_3_	22.1, CH_3_	21.3, CH_3_	21.8, CH_3_	21.7, CH_3_
17	22.3, CH_3_	22.4, CH_3_	22.1, CH_3_	22.3, CH_3_	22.7, CH_3_	22.8, CH_3_
18	19.4, CH_3_	17.8, CH_3_	16.8, CH_3_	16.9, CH_3_	17.4, CH_3_	17.4, CH_3_
19	167.0, C	166.9, C	168.1, C	167.9, C	168.5, C	168.6, C
20	17.2, CH_3_	12.0, CH_3_	12.0, CH_3_	13.3, CH_3_	12.4, CH_3_	13.1, CH_3_
OAc	170.2, C	170.2, C				170.4, C
	21.4, CH_3_	21.4, CH_3_				21.6, CH_3_
OMe	52.1, CH_3_	52.0, CH_3_	51.4, CH_3_	51.4, CH_3_	51.3, CH_3_	51.3, CH_3_

^a^ Recorded in CDCl_3_ at 150 MHz.

**Table 3 marinedrugs-23-00170-t003:** MIC values (μg/mL) of **1**–**9** and standard antibiotics against *Streptococcus parauberis*.

Compound	*Streptococcus parauberis* FP KSP28
**1**	>39.0
**2**	38.8
**3**	>33.0
**4**	>34.6
**5**	>33.2
**6**	>39.7
**7**	>33.2
**8**	37.4
**9**	31.6
Tetracycline ^a^	3.056
Ampicillin ^a^	4.642

^a^ positive controls.

## Data Availability

The data that support the findings of this study are available in the Appendix A of this article.

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
