# Peer review of "Antibacterial Methyl Ester Cembranoids from the Soft Coral Sarcophyton ehrenbergi and Their Structural Elucidation"

_marinedrugs, 2025, doi:10.3390/md23040170_

Round 1
Reviewer 1 Report
Comments and Suggestions for Authors
The manuscript by Yue-Wei Guo et al describes the isolation of six new and three known cembranoids from the soft coral Sarcophyton ehrenbergi. Biological evaluation of these compounds revealed that they display very modest antibacterial activity against Streptococcus parauberis.
Structurally, all six newly identified cembranoids (1-6) are very similar to previously reported compounds 7-9, sharing the same carbon skeleton but having minor differences in terms of substitution pattern.
Considering the lack of structural novelty and/or significant biological properties of any of the newly isolated compounds, I find this manuscript borderline for publication in Marine Drugs.
Additional Comments:
- How where compounds 7-9 characterized ?
- Also, no experimental details are provided on the isolation of 7 and 9.
- Abstract: “Their structures and absolute configurations were unambiguously established in the light of extensive spectroscopic data analysis, X-ray diffraction analysis, chemical conversion, and quantum mechanical calculations.” I did not see any chemical transformations in the manuscript. No obvious quantum mechanical calculations either (unless somehow the authors consider TDDFT-ECD as such)
- Page 8, line 222: Replace “but also offer a stable and reliable research protocols” by “but also offer reliable research protocols”
Author Response
Response to Reviewer #1
The manuscript by Yue-Wei Guo et al describes the isolation of six new and three known cembranoids from the soft coral Sarcophyton ehrenbergi. Biological evaluation of these compounds revealed that they display very modest antibacterial activity against Streptococcus parauberis.
Structurally, all six newly identified cembranoids (1-6) are very similar to previously reported compounds 7-9, sharing the same carbon skeleton but having minor differences in terms of substitution pattern.
Considering the lack of structural novelty and/or significant biological properties of any of the newly isolated compounds, I find this manuscript borderline for publication in Marine Drugs.
Response: We highly appreciate the reviewer’s valuable comments.
Additional Comments:
- How where compounds 7-9 characterized ?
Response: Thank you for the detailed review. The structures of three known related analogues 7–9 were characterized by comparing the 1H and 13C NMR data with that reported in references 10 and 12. The original 1H and 13C NMR spectra of compounds 7-9 (Fig S54-59) have been added in the revised version of Supporting Information.
- Also, no experimental details are provided on the isolation of 7 and 9.
Response: According to the reviewer’s kind suggestion, the experimental details of 7 and 9 have been added to the Experimental Section (page 8, line 255) and the first paragraph of Result and Discussion section in the revised manuscript.
- Abstract: “Their structures and absolute configurations were unambiguously established in the light of extensive spectroscopic data analysis, X-ray diffraction analysis, chemical conversion, and quantum mechanical calculations.” I did not see any chemical transformations in the manuscript. No obvious quantum mechanical calculations either (unless somehow the authors consider TDDFT-ECD as such).
Response: We appreciate the reviewer’s professional comment. In order to confirm the structural relationship between compounds 5 and 6, an acetylation reaction was carried out on 5. This chemical transformation was described in page 7, lines 193-197. In addition, the detailed acetylation experimental details of this chemical transformation have been added to the Materials and Methods Section in the revised manuscript. Furthermore, the NMR spectrum of the acetylated derivative of 5 (Fig. S60) has been also added in the revised version of Supporting Information. Finally, to avoid misunderstandings, the ‘quantum mechanical calculations’ has been replaced by ‘TDDFT-ECD calculations’ in the Abstract of the revised manuscript.
- Page 8, line 222: Replace “but also offer a stable and reliable research protocols” by “but also offer reliable research protocols”
Response: Following the reviewer’s kind suggestion, the statement “but also offer a stable and reliable research protocols” has been revised as “but also offer reliable research protocols” in the revised manuscript.
Reviewer 2 Report
Comments and Suggestions for Authors
The paper entitled “Antibacterial methyl ester cembranoids from the soft coral Sarcophyton ehrenbergi and their structural elucidation” describes the isolation of six new cembranoids from the soft coral Sarcophyton ehrenbergi in the South China Sea. The paper is well written, actual, presents the new structures of cembranoids of marine origin and could be accepted to Marine Drugs after revision. The topic of this manuscript is original and possesses novelty since the search of new the natural compounds with the promising biological activity, especially with antibacterial effect, is important. The main question addressed by this research is the discovery of new biologically active structures from the renewable raw materials. The conclusions consistent with the evidence and are detailed, the necessary arguments are presented and addressed to the main question posed. The figures and tables are concisely and clearly reflect the details of the discussed topic. The recommendations to the authors are the following: 1) the results and discussion part should be started from the description of isolation of new cembranoids and then – the characteristics of their NMR spectra and other methods of structure elucidation should be followed; 2) what was changed in the isolation process that gave the possibility to isolate new cembranoids? 3) please combine Fig 4 and 5 as one Fig; 4) please take attention that there are sections 2. Results and Discussion and 3. Discussion; 5) the marine material was collected in 2007 – please explain why only in 2025 the publication was submitted; 6) please compare the data of antibacterial activity with the literature data for other cembranoids.
Comments on the Quality of English Language
The English could be improved to more clearly express the research.
Author Response
Response to Reviewer #2
The paper entitled “Antibacterial methyl ester cembranoids from the soft coral Sarcophyton ehrenbergi and their structural elucidation” describes the isolation of six new cembranoids from the soft coral Sarcophyton ehrenbergi in the South China Sea. The paper is well written, actual, presents the new structures of cembranoids of marine origin and could be accepted to Marine Drugs after revision. The topic of this manuscript is original and possesses novelty since the search of new the natural compounds with the promising biological activity, especially with antibacterial effect, is important. The main question addressed by this research is the discovery of new biologically active structures from the renewable raw materials. The conclusions consistent with the evidence and are detailed, the necessary arguments are presented and addressed to the main question posed. The figures and tables are concisely and clearly reflect the details of the discussed topic.
Response: We are grateful for the reviewer’s kind and positive judgement.
The recommendations to the authors are the following:
1) the results and discussion part should be started from the description of isolation of new cembranoids and then – the characteristics of their NMR spectra and other methods of structure elucidation should be followed;
Response: Thanks for the reviewer’s kind suggestion. According to the reviewer's comments, the description of isolation process of the new cembranoids has been added before the structural characterization of the compound in the Results and Discussion section.
2) what was changed in the isolation process that gave the possibility to isolate new cembranoids?
Response: Thank you for your kind comment. The compounds described in this work all have methoxy groups and exhibit a characteristic methyl single peak of around 4.0 ppm on the 1H NMR spectra. Moreover, these compounds exhibit absorption at a wavelength of 254nm in UV light on TLC plates. By heating after spraying with anisaldehyde H2SO4 reagent, those compounds appear as purple spots which are different from the yellow spots of compounds with a five membered lactone ring that reported in previous work (Ref. 11). Those above observations can guide us to effectively isolate target compounds. In addition, as shown in the experimental description of the compounds’ separation process, the new cembranoids are mainly concentrated in the middle polar component Fra. D, while known compounds such as 7 and 9 are concentrated in the small polarity Fra. B.
3) please combine Fig 4 and 5 as one Fig;
Response: Thanks for the kind suggestion. In accordance with the reviewer’s suggestion, the Figures 4 and 5 have been combined as one Figure in the revised manuscript.
4) please take attention that there are sections 2. Results and Discussion and 3. Discussion;
Response: Thank you for your careful review. The repeated Section 3 Discussion has been removed.
5) the marine material was collected in 2007 – please explain why only in 2025 the publication was submitted;
Response: We highly appreciate the reviewer’s kind comment. The marine material, specifically coral samples in this case, was collected in 2007. We hold a profound respect for the preciousness of corals and value every sampling opportunity as well as each individual sample. Upon collection, all samples are stored in a refrigerator at -20 ℃ to safeguard the physiological characteristics of the metabolites within the coral samples. In addition, the species identification of coral samples and the chemical investigation conducted on the coral samples are time-consuming. It was not until 2023 that we initiated the systematic isolation and identification of this soft coral sample.
6) please compare the data of antibacterial activity with the literature data for other cembranoids.
Response: We appreciate the reviewer’s kind suggestion. In antibacterial bioassays against eighteen human and marine fish pathogens, compounds 2, 8 and 9 exhibited moderate antibacterial activity against Streptococcus parauberis with the MIC value of 38.8, 37.4 and 31.6 μg/mL, respectively, which are comparable to those of three previously reported cembranoids (lobocaloid B, 8.7 μg/mL; 11,12-epoxy-1E,3E,7E-cembratrien-15-ol, 30.4 μg/mL; sarcophytrol L, 32.2 μg/mL) [14].
[14] Zhu, S.-H.; Chang, Y.-M.; Su, M.-Z.; Yao, L.-G.; Li, S.-W.; Wang, H.; Guo, Y.-W. Nine new antibacterial diterpenes and steroids from the South China Sea soft coral Lobophytum catalai Tixier-Durivault. Mar. Drugs 2024, 22, 50. https://doi.org/10.3390/md22010050
Round 2
Reviewer 1 Report
Comments and Suggestions for Authors
The revised form of the manuscript is acceptable for publication.